# Decoding Metabolic Symbiosis between Pancreatic Cancer Cells and Cancer-Associated Fibroblasts Using Cultured Tumor Microenvironment

**DOI:** 10.3390/ijms241311015

**Published:** 2023-07-03

**Authors:** Yuma Nihashi, Xiaoyu Song, Masamichi Yamamoto, Daiki Setoyama, Yasuyuki S. Kida

**Affiliations:** 1Cellular and Molecular Biotechnology Research Institute, National Institute of Advanced Industrial Science and Technology (AIST), Tsukuba 305-8565, Japan; y-nihashi@aist.go.jp (Y.N.); son.xiaoyu93@aist.go.jp (X.S.); 2Tsukuba Life Science Innovation Program (T-LSI), School of Comprehensive Human Sciences, University of Tsukuba, Tsukuba 305-8572, Japan; 3Department of Research Promotion and Management, National Cerebral and Cardiovascular Center, Kishibe-Shimmachi, Suita 564-8565, Japan; myamamoto@ncvc.go.jp; 4Department of Clinical Chemistry and Laboratory Medicine, Kyushu University Hospital, Fukuoka 812-8582, Japan; setoyama.daiki.753@m.kyushu-u.ac.jp; 5School of Integrative & Global Majors, University of Tsukuba, Tsukuba 305-8572, Japan

**Keywords:** pancreatic ductal adenocarcinoma (PDAC), cancer-associated fibroblasts (CAFs), metabolic reprogramming

## Abstract

Pancreatic ductal adenocarcinoma (PDAC) is a highly aggressive cancer with a poor prognosis, largely due to its unique tumor microenvironment (TME) and dense fibrotic stroma. Cancer-associated fibroblasts (CAFs) play a crucial role in promoting tumor growth and metastasis, contributing to the metabolic adaptation of PDAC cells. However, the metabolic interactions between PDAC cells and CAFs are not well-understood. In this study, an in vitro co-culture model was used to investigate these metabolic interactions. Metabolomic analysis was performed under monoculture conditions of Capan−1 PDAC cells and CAF precursor cells, as well as co-culture conditions of PDAC cells and differentiated inflammatory CAF (iCAF). Co-cultured Capan−1 cells displayed significant metabolic changes, such as increased 2-oxoglutaric acid and lauric acid and decreased amino acids. The metabolic profiles of co-cultured Capan−1 and CAFs revealed differences in intracellular metabolites. Analysis of extracellular metabolites in the culture supernatant showed distinct differences between Capan−1 and CAF precursors, with the co-culture supernatant exhibiting the most significant changes. A comparison of the culture supernatants of Capan−1 and CAF precursors revealed different metabolic processes while co-culturing the two cell types demonstrated potential metabolic interactions. In conclusion, this study emphasizes the importance of metabolic interactions between cancer cells and CAFs in tumor progression and highlights the role of TME in metabolic reprogramming.

## 1. Introduction

Pancreatic ductal adenocarcinoma (PDAC) is a highly aggressive and lethal cancer with a poor prognosis. Despite significant efforts to improve therapeutic outcomes, the five-year survival rate for PDAC remains less than 10% [1]. The unique tumor microenvironment (TME) of PDAC, characterized by a dense fibrotic stroma [2,3,4] and low oxygen tension [5,6], plays a critical role in disease progression and treatment resistance [7,8,9]. Cancer-associated fibroblasts (CAFs) are a type of stromal cells that are found in the TME of many types of cancers, including PDAC [10,11,12,13]. They play a critical role in promoting tumor growth and metastasis through various mechanisms [14,15,16,17,18]. One way in which CAFs contribute to cancer progression is by altering the metabolic properties of the TME [19,20,21,22]. CAFs can enhance the glycolytic activity of cancer cells by secreting lactate, which is used by cancer cells as a source of energy [23,24]. Additionally, CAFs can stimulate the uptake of glucose by cancer cells through the secretion of growth factors such as insulin-like growth factor 1 (IGF-1) and transforming growth factor beta (TGF-β) [19,25,26]. CAFs also strongly contribute to the formation of a dense fibrotic stroma in the TME. This stroma can limit the delivery of oxygen and nutrients to cancer cells, creating a hypoxic environment that can further enhance the metabolic adaptation of cancer cells to survive and proliferate [24,27].

Metabolic adaptations are essential for cancer cells to sustain their growth and survival in the harsh TME [28,29]. However, the metabolic heterogeneity of PDAC cells and CAFs and their metabolic interactions remain poorly understood. Metabolomic analysis is a comprehensive approach to studying the metabolic profile of cells or tissues, which involves the identification and quantification of small molecules, such as metabolites, in a biological sample [30,31,32,33]. In the context of cancer research, metabolomic analysis can provide valuable insights into the metabolic alterations that occur in cancer cells and their interactions with the TME [34]. We previously used Capillary Electrophoresis-Time Of Flight Mass Spectrometry (CE-TOFMS) to perform targeted and untargeted metabolomic analysis of the cells [24]. Targeted analysis involves the quantification of specific metabolites, while untargeted analysis aims to identify and quantify all metabolites present in a sample. The metabolomic analysis revealed distinct metabolic profiles of PDAC cells and CAFs, showing that the co-culture of these cells resulted in intracellular metabolic reprogramming. However, the TME consists not only of cell space but also extracellular space. To understand the role of cancer cells and CAFs in the TME, it is necessary to examine metabolites in the extracellular space. This allows us to better understand roles such as secretion and incorporation by cells.

In this study, we aimed to identify both intra- and extrametabolic alterations occurring in PDAC cells and investigate the metabolic interactions between PDAC cells and CAFs in a cultured TME. The importance of understanding the metabolic properties of PDAC cells and CAFs lies in their potential as therapeutic targets. Targeting the metabolic vulnerabilities of cancer cells has emerged as a promising strategy for cancer treatment. Therefore, our study provides insights into the metabolic heterogeneity of PDAC and the potential role of CAFs in regulating tumor metabolism.

## 2. Results

### 2.1. Intracellular Metabolic Reprogramming in PDAC Cells Co-Cultured with CAF Precursors

An in vitro co-culture model was previously constructed to explore the molecular interactions between PDAC cells and CAFs [35]. Human adipose-derived mesenchymal stem cells (AD-MSCs) [36], which can heterogeneously differentiate into CAF precursors both in vitro and in CDX mouse models [37], were utilized in the study as CAF precursors. We previously demonstrated that myoblastic CAF (myCAF) is induced upon direct co-culture of cancer cells and AD-MSCs, and inflammatory CAF (iCAF) is induced upon indirect co-culture of these cells [24,35]. In this study, indirect co-culture was performed to collect individual culture supernatants, which suggests that a significant portion of differentiated CAF precursors may exist as iCAFs. An intracellular metabolomic analysis was performed on PDAC-derived Capan−1 cells both before and after co-culturing with CAF precursors, as previously demonstrated. The intracellular metabolites of both monocultured Capan−1 cells and co-cultured Capan−1 cells were analyzed using metabolomic analysis following a 7-day co-culturing period. A total of 183 metabolites were successfully detected and analyzed using an orthogonal-partial-least-squares discrimination (OPLS-DA) model to identify differences between the two groups (n = 3 each). The co-cultured metabolite profile was significantly different from that of normal conditioned Capan−1, as shown in Figure 1A. Figure 1B displays a heatmap that focuses on the top 25 metabolites that underwent characteristic changes. Among these metabolites, eight exhibited increased values after co-cultivation. Notably, our data show increased levels of 2-oxoglutaric acid (α-ketoglutarate, Log2FC = 0.48), involved in the TCA cycle, and lauric acid (Log2FC = 0.76), metabolized via beta-oxidation to produce ATP, in co-cultured Capan−1 cells. On the other hand, levels of certain amino acids, specifically serine, glutamine, methionine, and alanine, showed decreased levels after co-culture, with respective log2 fold changes of −0.58, −0.37, −0.34, and −0.42. The alterations of these metabolites in co-cultured Capan−1 cells may suggest a potential activation of energy production within the cells. However, further investigations, such as the evaluation of activation or upregulation of enzymes involved in energy generation pathways, are needed to provide more convincing evidence.

### 2.2. Intracellular Metabolic Differences in Co-Cultured Capan−1 Cells and CAFs

It was hypothesized that there would be significant differences in intracellular metabolites between the two cell types. A comparison of the co-cultured Capan−1 and CAFs revealed differences. A total of 188 metabolites were successfully detected and analyzed using an OPLS-DA model to identify differences between the two groups (n = 3 each). According to the OPLS-DA analysis, there is a substantial change in the T-Score (75.5%) (Figure 2A). The Variable Important in Projection (VIP) score is a measure derived from a PLS-DA model, and it signifies the importance of each variable in the projection used to separate classes. Essentially, a high VIP score for a variable means it is significant in differentiating between the groups in our analysis. In our study, the VIP plot indicated that isocitric acid was higher in Capan−1 than in CAFs (Figure 2B). This suggests that Capan−1 cells retain more isocitric acid intracellularly.

### 2.3. Identifying Secreted and Consumed Extracellular Metabolites in Mono- and Co-Cultures of PDAC, CAF Precursors, and CAFs

Metabolomic analysis of the water-soluble metabolites was performed using the collected culture supernatants (Figure 3A). A total of 76 metabolites were successfully detected, and Partial Least Squares Discriminant Analysis (PLS-DA) was performed using these metabolites to explicitly show the differences between five groups (n = 6 each) (Figure 3B). Distinct differences were observed between the supernatants of monocultured Capan−1 and CAF precursor cells and DMEM, with the co-culture supernatant showing the greatest disparity from DMEM among all conditions. However, there were few overall differences in the metabolite profiles between the culture supernatants collected separately from the top and bottom of the co-culture, indicating that the medium during co-culture is prone to mixing. The metabolite profiles from culture supernatants of both Capan−1 cells and CAF precursors were distinctly different, indicating significant variation in the metabolites that the cells secreted and consumed from the medium. Figure 3C displayed a heatmap focusing on the top 25 metabolites that underwent characteristic changes. Among these metabolites, the top seven, including amino acids such as arginine, glutamine and lysine, were present in larger quantities in the DMEM. In contrast, under co-culture conditions, seven metabolites, including the essential amino acids such as valine, tryptophan, and histidine, showed significant depletion. Valine is an amino acid that is metabolized to succinyl-CoA, tryptophan is utilized for the conversion from pyruvic acid to oxaloacetic acid, and histidine becomes 2-oxoglutaric acid and enters the TCA cycle. The depletion of these essential amino acids suggests their potential consumption for the activation of the TCA cycle. Additionally, citric acid and isocitric acid were not detected in the DMEM and supernatant of Capan−1 monoculture, but their release was observed in the medium of CAF precursors and CAFs. These findings support the notion that metabolically active cancer cells consume citric acid and isocitric acid provided by CAFs as fuel for their proliferation. Recent studies have highlighted proline as a potential biomarker for distinguishing PDAC patients from healthy individuals [38]. Furthermore, proline has been shown to play a crucial role in supporting the survival of PDAC cells under nutrient-deprived conditions, contributing to energy generation through the TCA cycle metabolism [39]. Additionally, glutamine is supplied to the TCA cycle and is involved in proline production. PRODH1, also known as proline dehydrogenase 1, is an enzyme involved in proline metabolism, catalyzing the conversion of proline to delta-1-pyrroline-5-carboxylate. Studies have indicated its role in promoting the survival of colon tumor cells under nutrient stress through mechanisms such as autophagy and ATP production [40,41]. In our study, we observed the consumption of glutamine by Capan−1 cells, and proline expression was not detected in monoculture but increased under co-culture conditions. By comparing the metabolites present in the culture medium with those in the culture supernatant, the specific metabolites consumed or produced and released by the cells were identified.

### 2.4. Metabolic Alterations in Water-Soluble Metabolites of Capan−1 Cancer Cell Culture Supernatant

This section aims to investigate changes in water-soluble metabolites found in the culture supernatant of Capan−1 cancer cells. The culture supernatant of Capan−1 cells grown in DMEM was compared with that of DMEM alone. OPLS-DA analysis was performed using a total of 76 metabolites obtained from the two samples. The results showed significant differences between the culture supernatant of Capan−1 cells and DMEM (Figure 4A). The supernatant of Capan−1 cell culture exhibited particularly small variations in metabolite release and consumption, indicating a consistent pattern across all the culture plates. S-shape plot analysis revealed a significant increase in glutamic acid and lactic acid and a significant decrease in arginine, lysine, glutamine, and other amino acids in the Capan−1 cell culture supernatant (Figure 4B,C). Alterations of metabolite composition in the culture supernatant are a direct reflection of the metabolic activities within the cells. In our study, significant changes in certain metabolites, such as amino acids and lactic acid, imply abnormal metabolic behavior in cancer cells. Specifically, increased consumption of amino acids from the DMEM and elevated levels of lactic acid, a byproduct of enhanced glycolytic metabolism often seen in cancer cells (a phenomenon known as the Warburg effect), is indicative of metabolic abnormalities. These changes suggest a shift from normal cellular metabolism towards a more glycolytic phenotype which is characteristic of many types of cancer cells, including PDAC.

### 2.5. Metabolic Activity of CAF Precursors

This section explores the consumption and release of metabolites by AD-MSCs as CAF precursors. A total of 76 metabolites were identified in the culture supernatant of CAF precursors. These metabolites were compared with those in a culture medium. Six replicates of each assay were prepared, and OPLS-DA was used to examine the metabolic profiles. As shown in Figure 5A, the metabolite composition in the culture medium of CAF precursors underwent significant changes. The OPLS-DA analysis revealed a 38% change in T-Score, indicating substantial differences in the metabolic profiles between CAF precursors and culture medium. In comparison to the variability in metabolic profiles between samples of culture medium, the variability between culture plates of CAF precursors was smaller, suggesting a certain level of regulation in metabolite release and consumption. We noted that a significant portion of pyruvic acid, the final product of glycolysis, is consumed by CAF precursors (Figure 5B,C). Given the observed increases in the levels of citric acid and isocitric acid (products of the TCA cycle) in the culture supernatant of CAF precursors, it is likely that some of the consumed pyruvates are being converted to acetyl-CoA for the TCA cycle. However, we acknowledge that other metabolic pathways could also be involved, such as the conversion of pyruvic acid to lactic acid, which warrants further investigation.

### 2.6. Comparison of Metabolites in Culture Supernatants of Capan−1 and CAF Precursors

Next, a comparison was made between the culture supernatants of Capan−1 and CAF precursors to investigate the differences in metabolites present in the culture supernatants between different cell types. Capan−1 is an epithelial-derived cancer cell with a very fast proliferation rate, requiring a large amount of energy. On the other hand, CAF precursors, similar to fibroblasts, are mesenchymal stem cells. They have a certain degree of proliferative capacity and exhibit a balanced intracellular metabolism, as shown in Figure 4 and Figure 5. A total of 76 metabolites were identified in both culture supernatants and compared between samples. Six replicates were prepared for each assay, and OPLS-DA was used to examine their metabolic profiles. As shown in Figure 6A, the culture supernatant derived from Capan−1 cells exhibited less variation between samples, indicating a more homogeneous cell population. According to the OPLS-DA analysis, there is a large change in one orthogonal x-score component in CAF precursors (16.9% of the orthogonal variation), while the T-Score change between Capan−1 and CAF precursors shows an even more pronounced difference of 30.7%. The S-shape plot analysis and VIP plot revealed that the release of citric acid and isocitric acid, as shown in Figure 5, is significant in CAF precursors, while lactic acid is released more abundantly in Capan−1 (Figure 6B,C). As can be seen from the PCA analysis in Figure 3B, Capan−1 and CAF precursors shift to opposite axes when compared to the culture medium, suggesting that they undergo entirely different metabolic processes.

### 2.7. Investigation of Metabolic Changes in Co-Culture of Capan−1 and CAFs

Finally, an investigation was conducted to examine the changes in metabolites in the supernatant of the co-culture of Capan−1 and iCAFs. The goal was to provide insights into potential metabolic interactions between Capan−1 and iCAFs during co-culture. This information could reveal crucial details about their respective roles in tumor progression and the TME. DMEM served as a baseline for evaluating the metabolic changes occurring in the culture supernatants of both cell types when grown together. Interestingly, few differences were observed in most metabolites between the upper part cultured with Capan−1 and the lower part cultured with iCAFs, except for uridine (Figure 7A). This indicates that the metabolites are well mixed in the separated upper and lower parts of the membrane, which corroborates its usefulness as an in vitro TME model.

To further investigate the metabolic products in the TME, the metabolites obtained from both fractions of the Transwell were combined and compared with the culture medium as a single group (n = 12). OPLS-DA analysis was performed using a total of 76 metabolites obtained from the two groups. The results showed significant differences between the supernatant of the co-culture and DMEM (Figure 7B). Again, the metabolites contained in the culture supernatant recovered from co-culture were very similar. Figure 7C displayed a heatmap focusing on the top 25 metabolites that underwent characteristic changes. Approximately half of these metabolites were specific to each cell type. Among these metabolites, the top 11 were present in larger quantities in the co-cultured supernatant, including citric acid and isocitric acid. To investigate the changes in more detail, a VIP plot for the top 25 metabolites was displayed (Figure 7D). As a result, the co-culture supernatant contained higher levels of malic acid, citric acid, isocitric acid, succinic acid, alanine, glutamic acid, and lactic acid. Taking into account the results of the single culture analysis in Figure 4 and Figure 5, it is believed that glutamic acid and lactic acid are released from Capan−1 cells, while citric acid and isocitric acid are released from iCAFs. Conversely, other metabolites appeared to be specifically released after co-culture.

## 3. Discussion

### 3.1. Interpretation of Metabolic Reprogramming in PDAC Cells Co-Cultured with CAF Precursors

Our results have revealed that PDAC cells undergo significant metabolic reprogramming when co-cultured with CAF precursors, as evidenced by changes in intracellular metabolite profiles. PDAC cells heavily rely on this metabolic reprogramming, as improving fatty acid oxidation has been suggested to lead to decreased hypoxia-induced autophagy and increased cell death after chemotherapy [42]. Even in the presence of fully functional mitochondria, enhanced glycolysis can be induced by hypoxia or hypoxia-inducible factor 1 (HIF-1). It has been reported that CAF precursors initiate metabolic reprogramming to support the growth and metastasis of heterogeneous cancer cells, depending on oxidative phosphorylation (OXPHOS) or glycolysis, which can switch in response to drugs or microenvironmental stimuli [43]. In fact, the coexistence of increased glycolysis and OXPHOS has been observed in some cancer cells, and these metabolic phenotypes have been reported to switch reciprocally in response to drug treatment or microenvironmental stimuli [44]. For example, stromal cells such as pancreatic stellate cells secrete alanine, which is assimilated by PDAC cells to support glutamine and glucose metabolism [45,46].

Another mechanism involves CAF precursors providing fuel to PDAC cells [47]. Our findings revealed an increase in 2-oxoglutaric acid (α-ketoglutarate) and lauric acid levels in co-cultured Capan−1 cells. This suggests an upregulation of the TCA cycle and beta-oxidation, respectively, indicating a metabolic shift in cancer cells to enhance energy production in the presence of CAF precursors. It has been suggested that in proliferative cancer cells, the high rate of metabolic flux and the deregulation of metabolism contribute to maintaining low concentrations of citric acid, thereby promoting the Warburg effect [48,49,50].

The observed decrease In amino ac”d le’els in co-cultured Capan−1 cells is indicative of heightened metabolic consumption. These observations are aligned with the notion that cancer cells have a higher metabolic demand, which can lead to the depletion of essential nutrients such as amino acids. In the TME, cancer cells are known to interact with other cell types, such as CAFs, to compensate for these nutrient deficiencies. These interactions could allow the cancer cells to access necessary nutrients, enhancing their survival and growth despite the nutrient-poor conditions within the TME [51,52].

### 3.2. Implications of Metabolite Consumption and Release in Mono- and Co-Cultures of PDAC and CAF Precursors

The comprehensive analysis of water-soluble metabolites in the culture supernatants of mono- and co-cultures of PDAC and CAF precursors provided valuable insights into the metabolic interplay between these cell types. The distinct metabolite profiles observed in the culture supernatants of Capan−1 cells and CAF precursors suggest that different metabolic pathways are employed by each cell type to sustain their growth and survival. These findings have broad implications for understanding the complex interactions within the TME and how these interactions contribute to tumor progression. In monoculture, distinct metabolite profiles were observed, indicating different metabolic pathways utilized by PDAC cells and CAF precursors for their growth and survival. The presence of specific metabolites in larger quantities in the culture medium suggests active production or uptake by the cells. For instance, arginine was abundant in the culture medium, indicating its consumption by the cells. Moreover, the absence of certain metabolites, such as lactic acid, in the culture medium and their presence in the co-cultured cells or Capan−1 cells suggests their release by these cells. This indicates that the metabolic dynamics are altered in the presence of co-culture, leading to the production or consumption of specific metabolites. Understanding the specific metabolites consumed and released by different cell types in mono- and co-cultures provides valuable insights into their metabolic interactions. These findings contribute to our knowledge of the complex metabolic interplay within the TME and its impact on tumor progression. While the metabolomic analysis provided valuable insights into metabolic changes, it was limited to water-soluble metabolites. Additional studies could explore changes in lipid-soluble metabolites. While this study provides insights into the metabolic interplay between PDAC cells and CAF precursors, functional studies would be necessary to validate the implications of these interactions on cancer cell survival and growth. This study provides a snapshot of the metabolic interactions at one time point. A longitudinal study to track metabolic changes over time would provide a more comprehensive understanding of the dynamics of the TME.

### 3.3. Future Research Directions

The current study has generated novel insights into the metabolic reprogramming of PDAC cells in co-culture with CAF precursors, as well as the consumption and release of metabolites in mono- and co-cultures of PDAC and CAF precursors. Despite the benefits of the in vitro co-culture model used in this study, it may not fully mimic the complex and dynamic nature of the TME in vivo. The study focuses on Capan−1 cells and CAF precursors, which might not fully represent the heterogeneity of PDAC and CAF precursors observed in clinical settings. Studies involving other cell lines or patient-derived samples would complement these findings. In vivo studies or more sophisticated in vitro models such as organoids may provide additional insights. Further research is needed to fully elucidate the underlying molecular mechanisms governing these metabolic changes and their functional consequences for cancer progression. Future studies could investigate the specific signaling pathways involved in the metabolic reprogramming of PDAC cells and the potential for targeting these pathways as a therapeutic strategy. Additionally, it would be of interest to examine the metabolic interactions between PDAC cells and other stromal cell types within the TME to gain a more comprehensive understanding of the metabolic adaptations that drive tumor progression.

## 4. Materials and Methods

### 4.1. Cells and Culture Condition

The immortalized human AD-MSC cell line ASC52telo (ATCC SCRC-4000) and the human pancreatic cancer cell line Capan−1 (ATCC HTB-79) were utilized and cultured following previously established protocols [35,37]. Briefly, both cell lines were cultured in Dulbecco’s Modified Eagle’s Medium (DMEM; FUJIFILM Wako Pure Chemical Corp., Osaka, Japan) supplemented with 20% fetal bovine serum (FBS), 1% nonessential amino acids, and 1% streptomycin-penicillin at 37 °C in a humidified atmosphere with 5% CO_2_. Both cells were expanded and maintained within 20 passages under these culture conditions.

### 4.2. In Vitro Monoculture Assay

In monoculture conditions, 4 × 10^5^ AD-MSCs and 4 × 10^5^ Capan−1 cells were seeded in 6-well culture plates, respectively. The culturing conditions are the same as those described above for maintaining cells. Culture supernatants were collected on day 7, and 72 h prior to that, the culture medium was replaced. The collected culture supernatant was centrifuged (200× *g*) for 5 min at room temperature to remove cell debris, followed by immediate freezing and storage.

### 4.3. In Vitro Co-Culture Assay

In the co-culture condition, Transwell^®^ culture inserts (Transwell^®^ Permeable Supports 24 mm Insert, 6-well plates, constar#3450, Corning, NY, USA) were applied. The lower compartment of the insert was seeded with 4 × 10^5^ AD-MSCs, whereas the upper compartment of the transwell membrane was seeded with 4 × 10^5^ Capan−1 cells. The culture supernatant from both the upper and lower compartments was collected separately, and the subsequent steps of supernatant purification and preservation followed the same protocol as the monoculture assay.

### 4.4. Metabolite Extraction from the Culture Medium

In total, 20 μL of the medium were mixed with 180 μL of ice-cold methanol, vortexed vigorously, and centrifuged at 21,500× *g* for 10 min at 4 °C. The supernatant was collected and evaporated to dryness. After dissolved in 200 mL of 0.1% formic acid and further diluted 100-fold, the samples were subjected to liquid chromatography-mass spectrometry (LC-MS) analysis.

### 4.5. LC-MS Analysis

Primary metabolites, such as sugar phosphates, TCA-cycle metabolites, and amino acids, were analyzed using the LCMS-8060 instrument (Shimadzu, Japan). The sample was separated on a Discovery HS-F5-3 column (150 × 2.1 mm, 3 μm particle size, Sigma-Aldrich, St. Louis, MO, USA) with mobile phases consisting of solvent A (0.1% formic acid) and solvent B (0.1% formic acid in acetonitrile). The column oven temperature was 40 °C. The gradient elution program was as follows: a flow rate of 0.25 mL/min: 0–2 min, 0% B; 2–5 min, 0–25% B; 5–11 min, 25–35% B; 11–15 min, 35–95% B; 15–25 min, 95% B; 25.1–30 min, 0% B. The parameters for the heated electrospray ionization source (ESI) in negative/positive ion mode under multiple reaction monitoring (MRM) were as follows; drying gas flow rate, 10 L/min; nebulizer gas flow rate, 3 L/min; heating gas flow rate, 10 L/min; interface temperature, 300 °C; DL temperature, 250 °C; heat block temperature, 400 °C; CID gas, 270 kPa. Data processing was carried out using the LabSolutions LC-MS software (Ver: 5.118, Shimadzu, Japan).

### 4.6. Statistical Analysis and Visualization

The metabolite data were exported in CSV format and subsequently uploaded to the MetaboAnalyst^®^ platform (https://www.metaboanalyst.ca (accessed on 21 April 2023)) capable of processing and analyzing comprehensive metabolic profiles [53]. A data integrity check was performed by default, and data filtering was carried out based on the mean intensity value. A significance level of *p* < 0.05 was set, and normalization was performed using Pareto data scaling when comparing samples from two groups. OPLS-DA was conducted. The results of OPLS-DA were visualized using S-shape plots and VIP plots to identify metabolite markers contributing to the discrimination. When comparing three or more groups, normalization was performed using auto data scaling. PCA and cluster analysis (heatmap) were employed to compare discriminant metabolite expression. Additionally, the pathway analysis module in this platform utilized metabolite peak intensities as input data. The data were log-transformed, autoscaled and matched against the human KEGG database [54,55,56].

## 5. Conclusions

In conclusion, our study provided valuable insights into the metabolic symbiosis between PDAC cells and CAFs within a cultured TME. Our findings revealed that the metabolic heterogeneity of PDAC cells and CAFs plays a crucial role in tumor progression, with CAFs supporting the metabolic needs of cancer cells through various mechanisms. We demonstrated that the co-culture of PDAC cells and CAFs led to significant alterations in the metabolic profiles of both cell types, indicating a complex interplay between these cells in the TME. This metabolic reprogramming can promote cancer cell survival, growth, and resistance to treatment, thus contributing to the aggressiveness of PDAC. Our study highlights the potential of targeting the metabolic vulnerabilities of PDAC cells and their interactions with CAFs as a promising therapeutic strategy. Further research should focus on elucidating the precise mechanisms underlying the metabolic crosstalk between PDAC cells and CAFs and exploring novel approaches to disrupt this symbiosis for improved treatment outcomes. In light of these findings, future research directions may include investigations into the molecular pathways that regulate metabolic adaptations in PDAC and CAFs, as well as the development of novel therapeutics that specifically target these metabolic interactions within the TME.

## Figures and Tables

**Figure 1 ijms-24-11015-f001:**
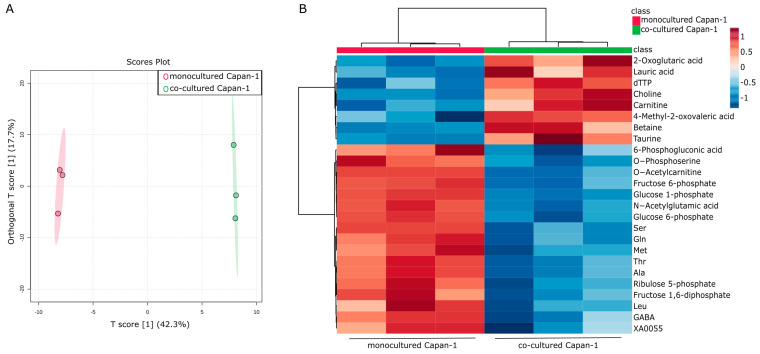
Intracellular metabolic reprogramming in PDAC cells co-cultured with CAF precursors. (**A**) Global trend of intracellular metabolite changes in cancer cells (Capan−1 cells). A supervised multivariate analysis, orthogonal-partial-least-squares-discriminant analysis (OPLS-DA), was performed using metabolomic datasets with Metaboanalyst 5.0 (https://www.metaboanalyst.ca/ (accessed on 21 April 2023)). The OPLS-DA score plot shows two groups of samples (n = 3 samples per group) based on the metabolomic datasets, with monocultured Capan−1 cells plotted in red and co-cultured Capan−1 cells in green. The model consists of one predictive x-score component: component t[1] and one orthogonal x-score component to[1]. The predictive variation in x (t[1]) represents the variation in the data that is correlated with the response variable, in this case, the difference between monocultured and co-cultured cells. This component accounts for 42.3% of the total variation. On the other hand, the orthogonal variation in x (to[1]) refers to the variation in the data that is not related to the response variable. Essentially, it captures the ‘noise’ or ‘unrelated variation’ in the data. In our model, it accounts for 17.7% of the total variation. (**B**) Heatmap comparing altered metabolites between monocultured Capan−1 cells (n = 3) and co-cultured Capan−1 cells (n = 3).

**Figure 2 ijms-24-11015-f002:**
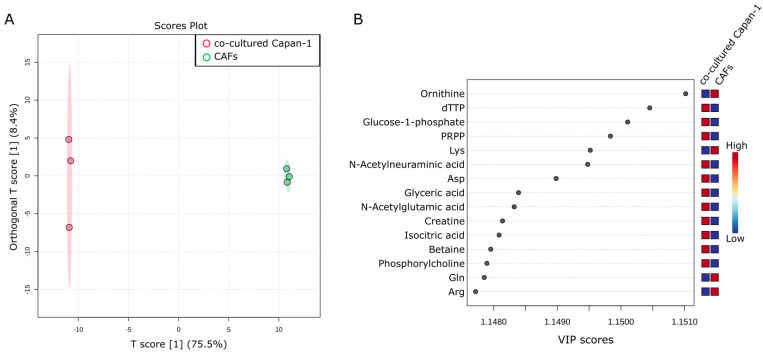
Intracellular metabolic differences in co-cultured Capan−1 Cells and CAFs. (**A**) Metabolites differentially detected in cells co-cultured with Capan−1 cells and CAFs. OPLS-DA was performed using metabolomic datasets with Metaboanalyst 5.0. The OPLS-DA score plot shows two groups of samples (n = 3 samples per group) based on the metabolomic datasets, co-cultured Capan−1 cells were plotted in red and co-cultured CAFs in green. The model consists of one predictive x-score component: component t[1] and one orthogonal x-score component to[1]. t[1] explains 75.5% of the predictive variation in x, to[1] explains 8.4% of the orthogonal variation in x. (**B**) A VIP plot corresponding to the score plot of OPLS-DA visualized the metabolite markers that importantly contributed to the discrimination.

**Figure 3 ijms-24-11015-f003:**
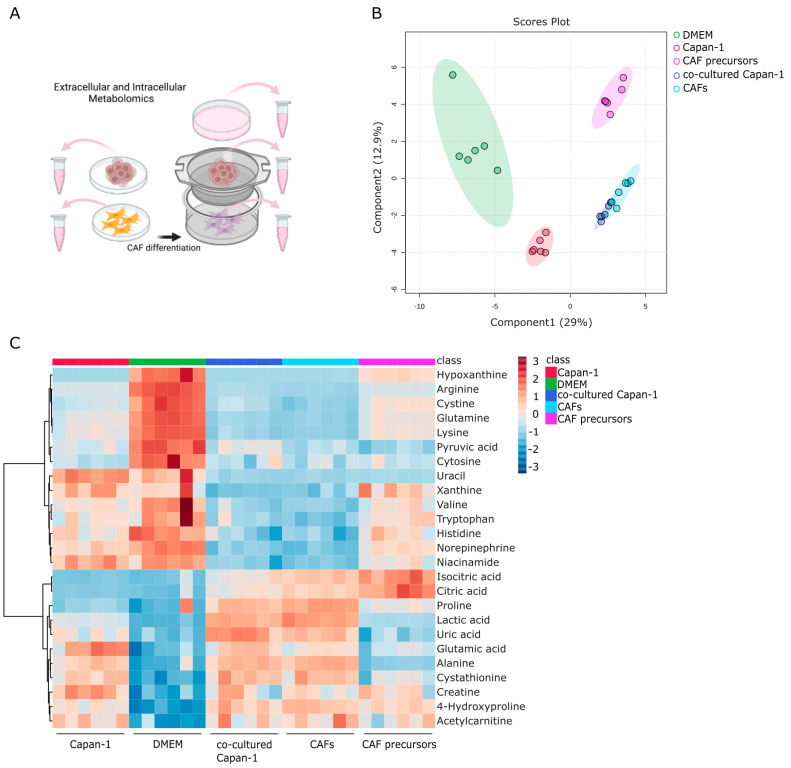
Identifying secreted and consumed metabolites in mono- and co-cultures of PDAC, CAF Precursors, and CAFs. (**A**) Global trend of extracellular metabolite changes in supernatant from Capan−1 cells, CAF precursors, and co-cultured cells. Schematic illustration of the extracellular metabolomic analysis to measure the metabolites in the culture medium. Supernatants were isolated from monoculture conditions or co-culture using a Transwell culture insert. (**B**) Principal component analysis (PCA) of extracellular metabolomic datasets of Capan−1 cells, CAF precursors, and mixed cells grown in each sole culture dish or a Transwell co-culture. (**C**) Heatmap comparing altered metabolites between supernatant from Capan−1 cells (n = 6), iCAF precursors (n = 6), co-cultured Capan−1 cells (n = 6), CAFs (n = 6), and DMEM (n = 6) as a baseline.

**Figure 4 ijms-24-11015-f004:**
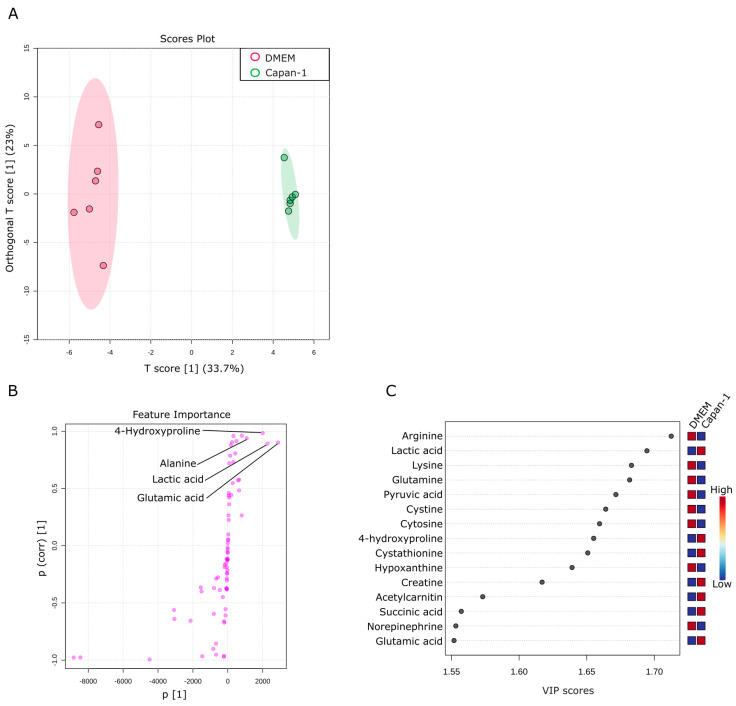
Metabolic alterations in water-soluble metabolites of Capan−1 cancer cell culture supernatant. (**A**) Metabolites specifically detected in the supernatants from sole Capan−1 cell culture. A supervised multivariate analysis, OPLS-DA, was performed. The OPLS-DA score plot shows two groups of samples (n = 6 samples per group) based on the metabolomic datasets; control DMEM media samples were plotted in red, and Capan−1 cell supernatants in green. The model consists of one predictive x-score component: component t[1] and one orthogonal x-score component to[1]. t[1] explains 33.7% of the predictive variation in x, and to[1] explains 23% of the orthogonal variation in x. (**B**) An S-shape plot corresponding to the score plot of OPLS-DA visualized the metabolite markers that contributed to the discrimination. (**C**) A VIP plot corresponding to the score plot of OPLS-DA visualized the metabolite markers that importantly contributed to the discrimination.

**Figure 5 ijms-24-11015-f005:**
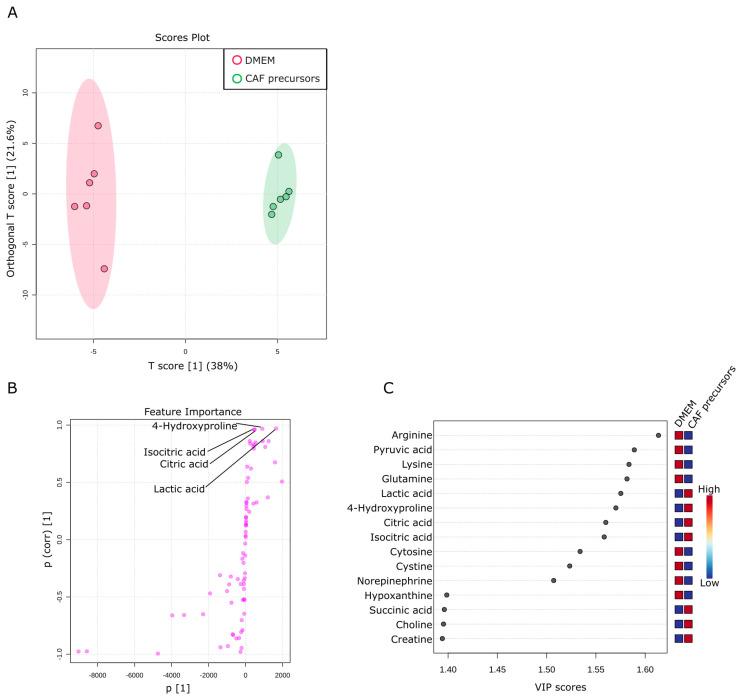
Metabolic activity of CAF precursors. (**A**) Metabolites specifically detected in the supernatants from sole CAF precursors culture. The OPLS-DA score plot shows two groups of samples (n = 6 samples per group) based on the metabolomic datasets; control DMEM media samples were plotted in red, and CAF precursors supernatants in green. The model consists of one predictive x-score component: component t[1] and one orthogonal x-score component to[1]. t[1] explains 38% of the predictive variation in x, to[1] explains 21.6% of the orthogonal variation in x. (**B**) An S-shape plot corresponding to the score plot of OPLS-DA visualized the metabolite markers that contributed to the discrimination. (**C**) A VIP plot corresponding to the score plot of OPLS-DA visualized the metabolite markers that importantly contributed to the discrimination.

**Figure 6 ijms-24-11015-f006:**
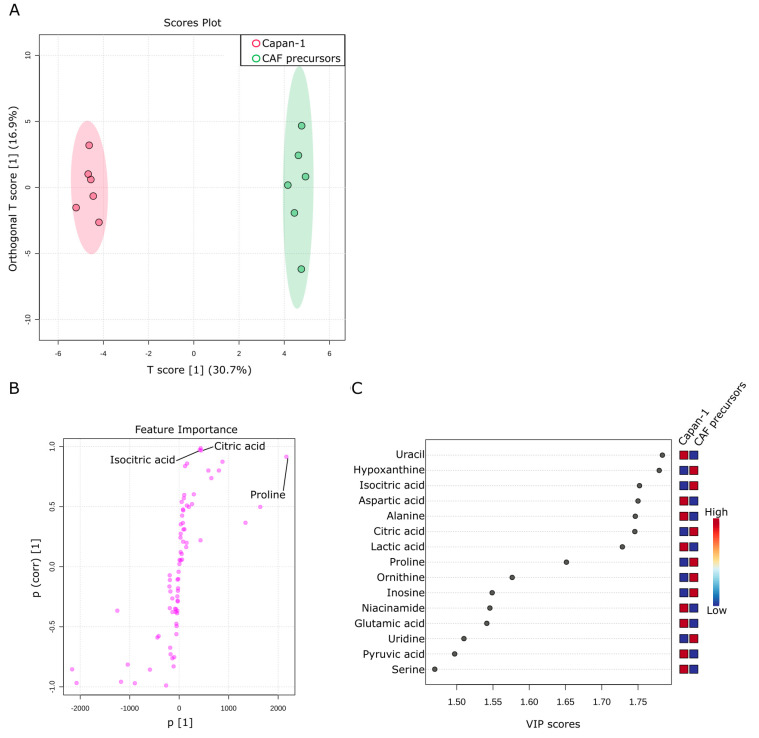
Comparison of metabolites in culture supernatants of Capan−1 and CAF precursors. (**A**) Metabolites differentially detected in the supernatants from sole Capan−1 cells and CAF precursors culture. OPLS-DA was performed using metabolomic datasets with Metaboanalyst 5.0. The OPLS-DA score plot shows two groups of samples (n = 6 samples per group) based on the metabolomic datasets: the supernatant of Capan−1 cells was plotted in red, and the supernatant of CAF precursors was plotted in green. The model consists of one predictive x-score component: component t[1] and one orthogonal x-score component to[1]. t[1] explains 30.7% of the predictive variation in x, and to[1] explains 16.9% of the orthogonal variation in x. (**B**) An S-shape plot corresponding to the score plot of OPLS-DA visualized the metabolite markers that contributed to the discrimination. (**C**) A VIP plot corresponding to the score plot of OPLS-DA visualized the metabolite markers that importantly contributed to the discrimination.

**Figure 7 ijms-24-11015-f007:**
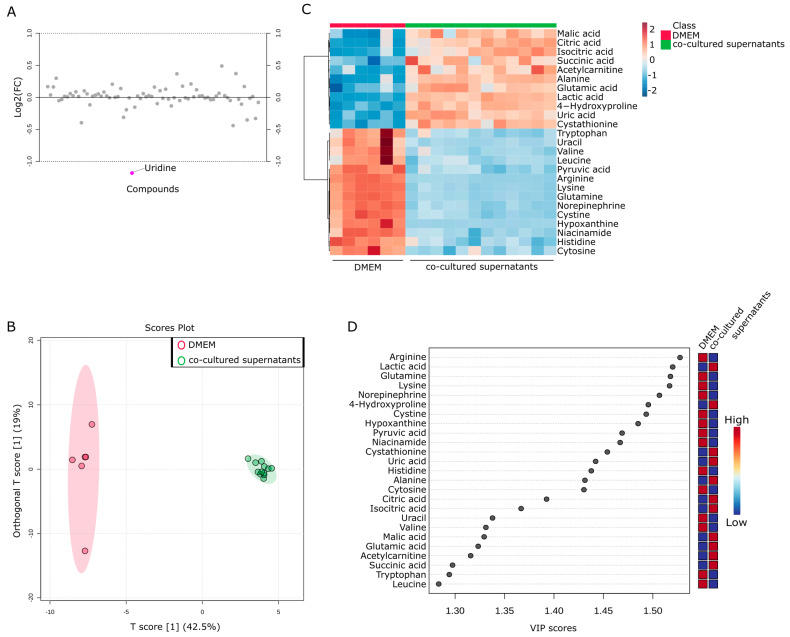
Metabolic changes in co-culture of Capan−1 and CAFs. (**A**) Metabolites differentially detected in the supernatants from co-cultured cells. Fold change analysis of metabolomic datasets of supernatants from co-cultured Capan−1 cells and CAFs. (**B**) Metabolites differentially detected in the supernatant of co-cultured and DMEM. The OPLS-DA score plot shows two groups of samples (DMEM media; n = 6, co-cultured supernatants; n = 12) based on the metabolomic datasets; control DMEM media samples were plotted in red, and co-cultured supernatant in green. The model consists of one predictive x-score component: component t[1] and one orthogonal x-score component to[1]. t[1] explains 42.5% of the predictive variation in x, and to[1] explains 19% of the orthogonal variation in x. (**C**) Heatmap comparing altered metabolites between DMEM media and co-cultured supernatants from the upper and lower part of the Transwell dish. (**D**) A VIP plot corresponding to the score plot of OPLS-DA visualized the metabolite markers that importantly contributed to the discrimination between DMEM media and co-cultured supernatants.

## Data Availability

All other data supporting the findings of this study are available from the corresponding authors upon request.

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
