# Peer review of "Decoding Metabolic Symbiosis between Pancreatic Cancer Cells and Cancer-Associated Fibroblasts Using Cultured Tumor Microenvironment"

_ijms, 2023, doi:10.3390/ijms241311015_

Round 1
Reviewer 1 Report
In the manuscript “Decoding metabolic symbiosis between pancreatic cancer cells and cancer-associated fibroblasts using cultured tumor microenvironment”, Yuma Nihashi and coauthors reported their characterization of the intra- and extracellular metabolic changes of pancreatic ductal adenocarcinoma (PDAC) cells upon incubation with cancer-associated fibroblasts (CAFs) in an in vitro tumor microenvironment (TME) setting.
Please take note of the following points that should be rectified:
Graphical Abstract
- Typo error “Extra cellula”.
- Schematics used should be redesigned, with addition of clearer figure caption to allow readers to better understand the flow of experiments conducted.
Results
- Line 100: Provide numerical quantification of the fold change in level of metabolites.
- Figure 1B: Figure caption states that “Capan-1 cell were plotted in red, and co-cultured Capan-1 cells in green”, but it states otherwise in the figure legend (red for co-cultured Capan-1 and green for mono-cultured Capan-1). Please rectify this error and update line 97-102 if necessary.
- Line 103: Alteration of metabolites level may not provide convincing proof that there is an activation of energy production. Perhaps authors may consider supplementing statement with additional evidence (e.g., activation or upregulation of enzymes involved in energy generation pathways)
- Line 114 – 115: Please explain what predictive and orthogonal variation in x represents.
- Line 122: It may be good to briefly elaborate how to interpret VIP Score Plot results before linking it to the suggested explanation.
- Line 140 – 145: Description of experimental set-up can be placed under Materials and Methods instead.
- Line 151: Please be consistent with the use of “fresh medium” and “culture medium”.
- Line 152 – 155: Consider rephrasing of sentence to make it more concise.
- Line 157 – 161: Consider rephrasing sentence to make it more coherent. Also, explain what it means by “characterized by tryptophan”.
- Line 166: Do not need to use full form of PDAC since it was abbreviated in previous section.
- Line 171: Please explain what is PRODH1.
- Line 201 – 203: Please elaborate on how changes in metabolites observed in culture supernatant reflects metabolic abnormalities.
- Line 229 – 230: Please provide additional evidence or explanation supporting the statement that pyruvate is converted to acetyl-CoA, since the latter is not the only possible downstream catabolite of pyruvate.
Discussion:
- Line 346 – 349: Please rephrase sentences to better explain the observation of decreased amino acid in co-cultured Capan-1 cells and how it can be linked to cancer cells compensating for the deficiency through interactions within TME.
- Figure 8 should be shifted to Results section.
Need moderate editing of English
Reviewer 2 Report
The manuscript by Nihashi and co-authors describes a thorough comparison of metabolomic difference in cancer metabolism induced by an important tumor microenvironment (TME) component. Cancer associated fibroblasts were grown in transwell co-culture to simulate one aspect of the TME. The findings reveal both anabolic and catabolic differences between the monocultured cancer cells (or TME cells) and the co-cultured cancer + TME cells, revealing some cancer metabolism insights.
First of all, the manuscript is extremely well-written. The work is describe consisely and appropriately for the methods used. Just one error on the grammatical side of things is a misspelling of “glutatic acid,” which should be fixed to “glutaric acid.”
The results sections contain intracellular metabolite comparison of the pancreatic cancer cell line Capan-1 in co-culture with cancer-associated fibrobast (CAF) models vs. monoculture. This work complements a previous study, where the CAF metabolites had be previously compared, as a demonstration of this method. The comparison methods look to me to be appropriate for making these comparisons, with at least 3 replicates per condition and more commonly n = 6.
Then, the authors proceed to compare the extracellular metabolomes of each cell type individually and in co-culture. Extensive data analysis is also performed, revealing key upregulated and downregulated metabolites when compared to fresh media as a baseline. Again, the analysis type appears to be appropriate, with n = 6 – 12 replicates.
This work reveals interesting trends in both anabolic and catabolic processes between the cells. Though the study is an in vitro study, the authors clearly describe these limitations and suggest several promising future applications for this work in more complex models of this cancer-TME interactions.
The materials & methods section is also clear, concise, and well-described. For these reasons I suggest publishing in IJMS.
Round 2
Reviewer 1 Report
All the concerns are well addressed. I recommend the manuscript for publication.
Author Response
Thank you for reviewing the revised manuscript and providing positive feedback. We greatly appreciate your recommendation for publication.